# Effect of Fermentation Duration on the Quality Changes of Godulbaegi Kimchi

**DOI:** 10.3390/foods11071020

**Published:** 2022-03-31

**Authors:** Jung-Min Park, Bo-Zheng Zhang, Jin-Man Kim

**Affiliations:** Department of Food Marketing and Safety, Konkuk University, Seoul 05029, Korea; slurpee24@naver.com (B.-Z.Z.); jinmkim@konkuk.ac.kr (J.-M.K.)

**Keywords:** Godulbaegi kimchi, antioxidant activity, antimicrobial activity, kimchi quality

## Abstract

Fermentative and antioxidative characteristics of Godulbaegi kimchi (LGK), a traditional, fermented Korean food, were conducted. For the study, LGK kimchi was made of Godulbaegi kimchi with pepper powder, salted shrimp, refined salt, green onions, and so on, and fermented at 5C for 6 months. The pH was decreased, and total acidity was increased during fermentation. Furthermore, lactic acid bacteria and yeast were increased, while the total viable count was decreased. The LGK showed the highest DPPH-scavenging activity, phenol content, and nitrite-scavenging activity with methanol extract among methanol, ethanol, and water. In addition, we screened strains among LGK kimchi with high antimicrobial activity and isolated them. We tested antimicrobial activity for 20 lactic acid bacteria, and we separated and identified nine strains of lactic acid bacteria with high antimicrobial activity. Given these results, LGK is expected to be an effective food in considerable antioxidative activity with an antimicrobial effect. These results are expected to serve as basic data for the study of Godulbaegi kimchi.

## 1. Introduction

Recently, natural products have become increasingly popular in the prevention of various diseases. Particularly, the anti-cancer properties of natural compounds are of interest, with research focusing on the discovery of anti-cancer and immunity-boosting substances. Moreover, studies have investigated the antioxidant activity of natural substances and their extraction process [1,2,3,4,5,6].

In the southern province of Korea, *Ixeris sonchifolia* Hance is a wild vegetable with a strong bitter taste, known as Godulbaegi (Korean lettuce) and belonging to the dandelion genus (*Taraxacum*) of the Asteraceae (also termed Compositae) family. Various types of Godulbaegi are grown in the mountains and fields of Korea [7]. There are a total of nine types, including *I. sonchifolia*, *Ixeris denticulate* (Hottu), and *Lactuca indica* L. var. *laciniata*. *I. sonchifolia* is also widely distributed in the northeastern part of China. *I. sonchifolia* is also a folk medicine that has been used for many years in China to improve health [8]. 

Kimchi is a Korean fermented food made from cabbage, onions, red pepper powder, garlic, ginger, and vegetables. Kimchi is a rich source of functional ingredients, including antioxidants, such as vitamins, flavonoids, and diverse phenolic compounds, as well as abundant lactic acid bacteria (LAB) involved in a complex fermentation process. Kimchi also contains free sugars, minerals, amino acids, fatty acids, polyphenols, flavonoids, and triterpenes. The health benefits of kimchi include its ability to enhance intestinal health and prevent constipation as well as display anti-mutagenic and anticancer effects. *I. sonchifolia* is widely used in Korea to prepare kimchi pickles or kimchi. Godulbaegi kimchi has been mainly used in the southern region of Korea and has been established as a local food in the region. 

Godulbaegi is eaten as raw greens in spring or soaked in kimchi in autumn and has been used medicinally, as it is known for improving blood circulation and dissipating blood stasis to relieve blood stasis pain, among other effects [1,9,10,11]. Among the diverse varieties of kimchi, Godulbaegi (Korean lettuce, *I. sonchifolia*) kimchi, which is consumed as a delicacy in the southern province of Korea, contains high levels of polyphenols and dietary fiber [12].

Recently, Godulbaegi was associated with cardiovascular disease [13,14] and the mechanism of oxidative stress modulation of antioxidant capacity [15]. Currently, research on Godulbaegi is mainly aimed at its antioxidants [10,16] and antitumor compounds [17] and its component analysis by HPLC/MS. The efficacy of Godulbaegi is highlighted by the various compounds present in Godulbaegi, red pepper powder, garlic, and ginger, which are the main ingredients, as well as substances produced by LAB (Lactic acid bacteria) fermentation process [18]. The growth of LAB in Godulbaegi acts as a beneficial probiotic, causing the production of various substances [19]. Various antioxidant benefits have been reported for metabolites produced by LAB during fermentation, and thus, there is an increasing interest in Godulbaegi as a functional food [20]. Therefore, in this study, we comparatively analyzed the physicochemical properties, antioxidant activity, and antibacterial activity of Godulbaegi kimchi according to storage period. In addition, we investigated the potential of Godulbaegi kimchi as a healthy functional food material.

## 2. Materials and Methods

### 2.1. Material and Preparation of Godulbaegi Kimchi

The main material in the kimchi used in this experiment was Godulbaegi. Auxiliary materials consisted of dried red pepper powder, anchovy fish sauce, salted shrimp, refined salt, garlic, scallions, onions, ginger, carrots, green onions, and white sugar. All materials were produced in Korea, purchased from a large shopping mall (Emart, Seoul, Korea) in the Gwangjin District of Seoul and delivered to the laboratory within 30 min. Godulbaegi was soaked in 5% brine for 48 h to bring out the bitter taste, then washed 3 times under running water and drained for 30 min. The materials for kimchi were as follows: Godulbaegi (83.0%), dried red pepper powder (4.0%), anchovy fish sauce (3.5%), salted shrimp (2.3%), refined salt (2.3%), garlic (1.6%), scallions (1.4%), onions (0.5%), ginger (0.5%), carrots (0.4%), green onions (0.4%), and white sugar (0.1%). The dried ingredients were freeze-dried using a freeze dryer (Freeze Dryer-5, Ilsin Engineering, Co., Dongducheon, Gyeonggi, Korea), sealed and stored in a freezer maintained at −20 °C, and then ground and used whenever necessary. Subsequently, 2 kg of each were placed in a plastic container (25 × 15 × 20 cm) and stored at 5 °C, and samples were taken on the 6th month of storage. The samples taken were short-term fermented Godulbaegi kimchi (SGK, fermented for 7 days) and long-term fermented Godulbaegi kimchi (LGK, fermented for 6 months). Additionally, due to the acidic environment and the presence of many LAB during the fermentation process, kimchi is very durable [21]. Godulbaegi kimchi is generally eaten within six months of purchase. Therefore, we evaluated Godulbaegi kimchi after a week and after six months of storage.

### 2.2. Sampling, pH, and Total Acidity

Five grams of samples were added to 45 mL of sterile distilled water, blended with a stomacher (Stomacher ^®^ 400 circulator; Seward Inc., West Sussex, UK), and filtered using filter paper (Whatman, Kent, UK). The pH of the SGK and LGK solutions was measured using a pH meter (pH Basic+; Sartorius AG, Göttingen, Germany). A homogeneous solution was obtained and filtered using filter paper (Whatman), and 0.1 mol/L NaOH was used to neutralize the kimchi solution. Ten milliliters of NaOH were used for total lactic acid content conversion and to determine the acidity (%, *w*/*v*) of each solution by titration of SGK) and LGK.

### 2.3. Number of LAB, Total Bacteria, and Yeast

LAB, total bacteria, and yeast were incubated by homogenizing the initial fermented Godulbaegi kimchi and LGK individually. LAB was cultured using de Man, Rogosa, and Sharpe (MRS) agar (Difco, Detroit, MI, USA) and bromocresol purple agar (BCP) agar. Cultures were incubated under anaerobic conditions at 37 °C for 48 h [22]. Plate count agar (Eiken Chemical, Tokyo, Japan) was used for the total bacterial count, using the homogenized solution, and incubated at 30 °C for 72 h [23]. Yeast was counted using potato dextrose agar (Difco) at 25 °C for 5 days and identified by its shape and size [24].

### 2.4. Sample Preparation and Extraction Yield

Godulbaegi kimchi was separated and cut into small pieces and freeze dried using a freeze dryer (EYELA N-1000, Tokyo Rikakikai Co., Ltd., Tokyo, Japan) at −50 °C, 35 mm Hg. Samples were stored in an air-tight container at −20 °C prior to further use. The extracts were prepared according to the method described by Mohd-Esa et al. (2010), with modifications [25]. *I. sonchifolia* was extracted using either distilled water (J. T. Baker, NJ, USA), methanol (J. T. Baker), or ethanol (J. T. Baker) for 24 h at room temperature, using an orbital shaker (JeioTech, Daejeon, Korea).

The mixture was filtered through a filter paper (Whatman no. 4, WM1004090). The filtrate was considered to be Godulbaegi kimchi extract and used for the antioxidant activity assays.

The extract was vacuum concentrated in each solvent, and the dried extract was weighed. The extraction yield was calculated using the following equation:% Yield = (extract (g)/raw material (g, dry weight) × 100

### 2.5. Determination of Total Polyphenols

The total polyphenol contents in diverse sample extracts were determined using the Folin–Ciocalteu colorimetric method [26]. A 20 mL sample of each extract filtrate was mixed with 1.58 mL of water, and 100 μL of Folin–Ciocalteu’s phenol reagent was added to the mixture. Then, in a 3 min reaction, 300 μL of 20% (*w*/*v*) sodium carbonate solution was added, and the mixture was incubated for 30 min at 40 °C. The absorbance of each sample was determined with a spectrophotometer at 765 nm (Beckman du 530, Brea, CA, USA). The total polyphenol content was calculated as gallic acid equivalent (µg of GAEs/mg extract) and calibrated. The total polyphenol content was calculated using the following equation:Phenolic content = 0.031 × A sample + 0.159

### 2.6. Measurement of Free-Radical-Scavenging Activity

Free radical scavenging was evaluated using the 2,2-diphenyl-1-picrylhydrazyl (DPPH) free radical by Shimada et al. [27]. Four milliliters of each sample extract were added to 1.0 mL of 1 mM DPPH solution. Then, 20, 40, 60, 80, and 100 mg/mL samples were prepared by adding 1 mL of DPPH solution to 5 mL of 80% methanol. After a 30 min incubation at 25 °C, the absorbance at 517 nm (UV-1601, Shizuoka, Japan) was recorded. The inhibitory of DPPH was expressed by the following equation:DPPH free radical-scavenging activity (%) (1 − A sample/A blank) × 100

### 2.7. Nitrite Scavenging Activity

Nitrite scavenging was generated from sodium nitroprusside and was measured according to the method by Gray and Dugan [28]. The nitrite-scavenging activity was conducted under various range of pH 1.2, 3.0 and 6.0, respectively by measuring absorbance at 520 nm using 1 mL of sample extract and Griess reagent. This solution was incubated for 1 h at 37 °C and subsequently mixed with 5 mL of 2% acetic acid and 0.4 mL Greiss reagent (1% sulfonic acid in 30% acetic acid and 1% naphthylamine in 30% acetic acid) and kept at room temperature for 15 min. The absorbance of the chromophore formed during the diazotization of sulphanilamide and nitrite, and subsequent binding with napthylethylenediamine was measured in 520 nm. The activity of nitrite scavenging (%) was determined by the following equation:Nitrite scavenging activity (%) = [1 − (A − B)/C] × 100
A.The absorbance of 1 mM NaNO_2_ added sample after allowing to stand for 1 h;B.The absorbance of control;C.The absorbance of 1 mM NaNO_2_.

### 2.8. Antimicrobial Activity

After analyzing antioxidant activities, we selected LGK with excellent antioxidant activity and conducted an antimicrobial test. A total of 20 colonies from LGK, of different shapes and sizes, were selected and incubated at 37 °C for 48 h in MRS broth. *Escherichia coli* KCCM 21052, *Salmonella typhimurium* p99, *Staphylococcus aureus* KVCC BA1100335, and *Listeria monocytogenes* KVCC BA0001449) were used as indicators to determine the antibacterial activity of 20 separate strains isolated from LGK. The agar disc method was used to determine the antibacterial activity of compounds produced by each isolate against the aforementioned food-borne pathogens. Paper disks were impregnated with 50 μL of bacterial suspension containing approximately 10^7^ CFU/mL of LGK from isolated bacteria. The paper disks containing the bacterial suspension were placed on the plates and incubated at 37 °C for 48 h. The antibacterial activity was expressed according to determining the diameter of the clear zone of growth inhibition.

### 2.9. Identification of LAB from Godulbaegi Kimchi

Colonies with the highest antibacterial activity were selected for taxonomic identification. DNA was extracted using the Power-Prep DNA Extraction Kit (Kogene Biotech, Seoul, Korea). The extract of DNA was used for polymerase chain reaction (PCR) with the two primers 27F (5′-AGAGTTTGATCCTGGCTCAG-3′, forward) and 1492R (5′-GGCTACCTTGTTTACGACTT-3′, reverse). Subsequent identification was performed using the 16S rRNA gene sequence analysis provided by SolGent (SolGent Co., Ltd., Daejeon, Korea). The phylogenetic tree was determined with the neighbor-joining method with the Molecular Evolutionary Genetics Analysis (MEGA) 7 software (Available online: https://www.megasoftware.net/ accessed on 20 February 2022).

### 2.10. Statistical Analysis

All experiments were conducted in triplicates and expressed as mean ± standard error. Statistical analyses were determined in the SPSS program (SPSS version 12.0, SPSS Chicago, IL, USA) using unpaired *t*-tests repeated measures analysis of variance (ANOVA) when appropriate. If the data was statistically significant by ANOVA (*p* < 0.05), differences in the means were determined using Duncan’s multiple range tests.

## 3. Results and Discussion

### 3.1. pH and Acidity Value of Godulbaegi Kimchi

Usually, in the early stage of kimchi fermentation, the pH, acidity, and microbial composition change very actively [29]. These parameters can be used as indicators to judge the quality of kimchi while also affecting the flavor of kimchi. The pH and acidity value changes in SGK according to the fermentation period are shown in Table 1. Early on, the average pH of SGK was 4.43, and the acidity value ranged 0.46%. Then, at 4 °C, pH tended to decrease, and the acidity value increased as fermentation proceeded. Therefore, by 6 months, the average pH and acidity value of LGK had reached 4.20 and 0.92%, respectively. Various conditions such as the composition of subsidiary materials, salinity, and storage temperature affect the pH of matured kimchi [30]. In addition, our results showed that the increase in acidity of LGK was due to the production of organic acids, which also affect kimchi taste [20,31,32]. The pH results were similar to those of typical fermentation processes reported previously [33]. Changes in pH or total acidity content are commonly used as indicators of ripeness during Godulbaegi kimchi fermentation. A pH of approximately 4.2–4.4 and acidity values ranging between 0.5–0.75% are optimal conditions for Godulbaegi kimchi consumption [34,35].

### 3.2. Microbial Load Analysis

The changes in the LAB, total bacteria, and yeast during the storage of SGK and LGK are shown in Table 1. Changes in LAB counts during SGK fermentation showed a time-dependent increase from 6.06 log_10_ CFU/g to 6.59 log_10_ CFU/g. Other studies have shown that the number of LAB in kimchi increases rapidly during the early stages of fermentation until the peak is reached after approximately 8 days, then slowly decreases and gradually stabilizes [36]. Nonetheless, as fermentation progressed, the antibacterial substances secreted by certain LAB gradually increased [37]. This inhibited the growth of other bacteria, decreasing the total bacterial count from 5.85 log_10_ CFU/g to 4.66 log_10_ CFU/g. Conversely, as fermentation progressed, yeast counts gradually increased from 1.24 log_10_ CFU/g to 3.06 log_10_ CFU/g, indicating that the fermentation process of kimchi is suitable for yeast growth. In general, the fermentation pattern of kimchi is such that as fermentation progresses in the early stages, the microbial quantity increases. Fermentation then gradually decreases as microbial counts reach the maximum level. This process gives kimchi a unique taste and aroma, accompanied by biochemical changes.

### 3.3. Extraction Yield

Table 2 shows the extract yield of LGK using either methanol, ethanol, or water. The extraction yield varied in the following order depending on the solvent used for extraction and duration of fermentation, namely methanol extraction > ethanol extraction > water, extraction and ranged from 48.54% to 61.51%.

Methanol extracts have been reported to have higher antioxidant capacities. Our results were consistent with previous research, showing the following order: methanol > ethanol > water extracts [38].

### 3.4. Total Phenolic Content

The total phenolic content was conducted using the Folin–Ciocalteu method. The content of phenolic compounds was determined using a regression equation of the calibration curve (y = 0.031x + 0.1593, R2 = 0.9922) and expressed in GAE. Table 2 shows that the total phenolic content of LGK extracts in methanol, ethanol, or water was 38.02, 77.40, and 79.01 μg of GAE/mg extract, respectively. A significantly higher total phenol content was determined in samples that underwent higher fermentation periods. That is, phenol content increased with increasing duration of fermentation because phenolic acids, such as coumaric and ferulic acids, form ethyl or vinyl phenol derivatives through reactions with microorganisms. Generally, the total phenol content increases as fermentation proceeds, and this phenomenon was consistent with the results of this experiment [39].

Moreover, the methanol and ethanol Godulbaegi extracts had higher polyphenol content than the water extract. Therefore, methanol and ethanol were the most efficient solvents for extracting the antioxidants from the samples. Therefore, LGK is high in total polyphenols and is expected to play a role as an antioxidant.

### 3.5. DPPH Radical Scavenging

The results in Figure 1 show the scavenging activity on DPPH radicals of the different LGK extracts increased in a dose-dependent manner with DPPH doses varying between 20 and 100 mg/mL. The scavenging activity on DPPH radicals of LGK ranged from 17.56% to 88.37%. These variations may have been caused by differences either in potency or in the concentration of reducing substances, mainly phenolic compounds, which reflects the effect of red pepper seed on Godulbaegi kimchi antioxidant activity during fermentation. DPPH-radical-scavenging activity studies reported that fermented kimchi in different solvent extracts contains active materials, such as phenolic compounds, vitamin C, and phenolic acid, which can show scavenging activity on DPPH radicals. The scavenging activity on DPPH radicals of LGK was significantly higher (*p* < 0.05) in methanol ethanol extracts than in water extracts. Particularly, the DPPH-radical-scavenging activity of the LGK methanol extract ranged from 58.12% to 88.35% using DPPH doses between 20 and 100 mg/mL. These data highlight the antioxidant properties of Godulbaegi kimchi. Thus, Godulbaegi kimchi possesses antibacterial and antioxidant activities that could be useful for the development of various pharmaceutical and food products through future research. Aarti et al. [40] reported that the DPPH-radical-scavenging activity of *Lactobacillus brevis* LAP2 increased in a concentration-dependent manner (18.8–68.35% at 108–109 CFU/mL). Additionally, the scavenging activity on DPPH radicals of the extracts increased according to the duration of fermentation.

### 3.6. Nitrite Scavenging in Godulbaegi Kimchi

Table 3 shows the change in the quantity of nitrite scavenging depending on the time of fermentation and solvent used for extraction at various pH levels (pH 1.2, 3.0, and 6.0). The degree of degradation was higher at acidic pH conditions, being highest at pH 1.2 in all extraction solvents (*p* < 0.05). Furthermore, the result showed that the methanol and ethanol extracts had the highest nitrite-scavenging activity among the extracts, followed by the water extract. The nitrite-scavenging effect was highest at pH 1.2 (81.90%) for methanol extracts. This was likely because of the various phenolic substances in Godulbaegi kimchi. This result was similar to that of a previous study wherein *Lactobacillus sakei* was able to deplete nitrite and degrade N-nitrosodimethylamin (NDMA) in MRS broth, and LAB was able to deplete NaNO_2_. In addition, the inoculation of *L. sakei*, *L. curvatus*, and *L. brevis* into kimchi resulted in a marked reduction in the nitrite levels, which might be due to the nitrite-scavenger effects of LAB. Green onion, garlic, and dried red pepper contain a variety of active compounds, such as allyl sulfides, carotenes, phenolic compounds, and ascorbic acid, which prevent nitrosation [41,42].

### 3.7. Antimicrobial Activity of Lactic Acid Bacteria Isolates against Target Bacteria 

Preliminary screening of the antimicrobial activities of LAB were verified for 20 strains isolated from LGK. Among them, we selected nine strains with antibacterial effect and numbered them 1–9. The results in Table 4 show the LAB strains’ antimicrobial activities were effective against Gram-negative bacteria but not against Gram-positive ones. For instance, sample 1 had strong inhibition zones of >1.5 mm for *E. coli* (KCCM 21052) and of 1.2–1.5 mm for *S. typhimurium* P99. Samples 4, 6, and 9 had clear inhibition zones of over 1.0 mm for *E. coli* (KCCM 21052) and *S. typhimurium* P99. However, the strains had no inhibition against Gram-positive bacteria such as *S. aureus* KVCC BA1100335 and *L. monocytogenes* KVCC BA0001449. This may be because the cell walls of Gram-positive bacteria hamper the entry of antimicrobial compounds. A previous study showed that *L. sakei* have a weak antimicrobial effect on *L. innocua* and *S. aureus* [43]. In this study, we found that antimicrobial compounds produced during LGK fermentation by LAB have high antibacterial activity against two Gram-negative food-borne pathogens. Therefore, LAB isolated from LGK can be used to make natural preservatives, which are important for food safety and for the food distribution industry [44,45].

### 3.8. Identification by 16S rRNA Sequencing

In total, nine LAB strains were selected for 16S rRNA sequencing based on their antimicrobial activity. 16S rRNA sequencing revealed that strain 4 showed 100% homology with *L. sakei* NBRC 15893. Strain 9 showed 99.09% homology with *Lactobacillus graminis* G90. Sample strains 1, 2, 3, 5, 6, 7, and 8 exhibited more than 99.86% homology with *L. sakei* subsp. DSM 20017. Figure 2 shows the phylogenetic tree constructed with the neighbor-joining method and the 16S rRNA gene sequences of the strains. In conclusion, various types of LAB were present in LGK, but *L. sakei* and *L. graminis* were the main types exhibiting the antimicrobial activity found in the LGK isolates in our study. These strains showed similar morphological features, and both secrete antioxidant molecules. In particular, *L. sakei* had a key role in meat preservation and fermentation mainly producing antioxidants with antibacterial effects, similar to our results [46,47,48]. The isolated strains can be widely used in the manufacture of natural preservatives and as natural additives in the production of certain fermented foods. Moreover, we found that the fermentation period affected the extraction yield, DPPH scavenging, content of total phenol, and antioxidant activity of Godulbaegi kimchi. These data describe the basic aspects of Godulbaegi kimchi. Nonetheless, to establish the value of Godulbaegi kimchi, more diverse functional ingredients and various physiological functional comparative studies are needed.

## 4. Conclusions

This study was conducted to investigate microbial and antioxidative characteristics of Godulbaegi kimchi (LGK). The pH was decreased, and total acidity was increased during fermentation. The effects obtained from the extraction of methanol, ethanol, and water from fermented Godulbaegi kimchi on antioxidant activity were investigated. In DPPH scavenging activity, phenol content, and nitrite-scavenging activity, methanol extract in Godulbaegi kimchi has the highest antioxidant activity among ethanol and water. The results of this study suggested that fermented Godulbaegi kimchi has a high antioxidant content. Additionally, Lactic acid bacteria such as *Lactobacillus sakei* NBRC 15893, *Lactobacillus graminis*, and *Lactobacillus sake,* with antibacterial activity, were identified in fermented Godulbaegi kimchi, traditional Korean kimchi. However, to establish the value of Godulbaegi kimchi, it is judged that more diverse functional ingredients and various physiological functional comparative studies of Godulbaegi kimchi are needed in the future. 

## Figures and Tables

**Figure 1 foods-11-01020-f001:**
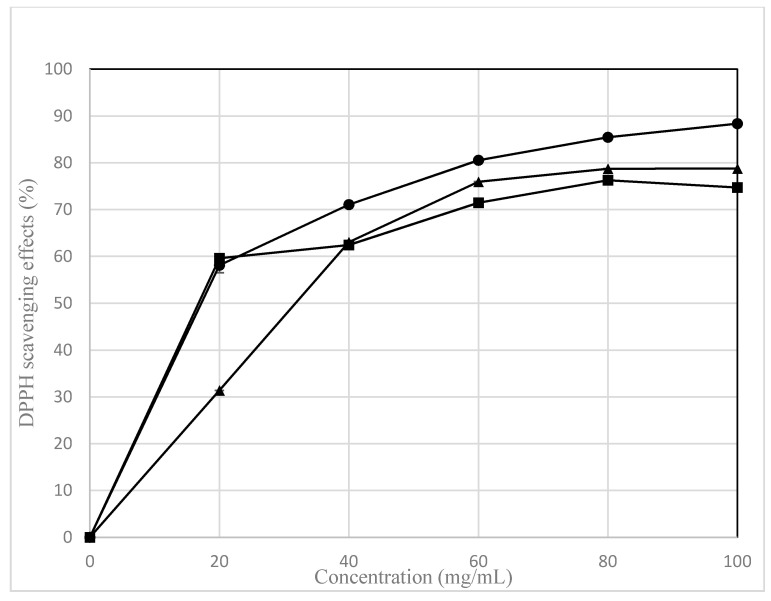
Scavenging activity on DPPH radicals of the methanol, ethanol, and water extracts in LGK (six months fermented Godulbaegi kimchi). The symbols express the following samples: closed circle, methanol extract; closed triangle, ethanol extract; closed square, water extract.

**Figure 2 foods-11-01020-f002:**
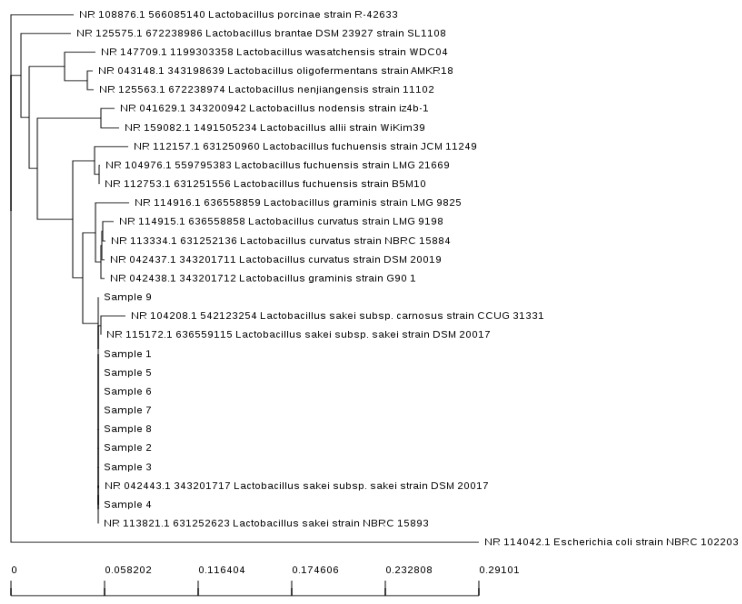
Phylogenetic tree construction using the neighbor-joining method and gene sequences, based upon 16S rRNA sequencing. The figure shows the positions of strains and other closely related lactic acid bacteria (LAB) isolated from long-term fermented Godulbaegi kimchi (LGK).

**Table 1 foods-11-01020-t001:** Changes in pH, acidity value, and numbers of LAB, total bacteria, and yeast.

	SGK ^a^	LGK ^b^
pH	4.43 ± 0.14 ^c^	4.20 ± 0.13
Total acidity (%)	0.46 ± 0.02	0.92 ± 0.03
Lactic acid bacteria (CFU/mL)	6.06 ± 0.28	6.59 ± 0.10
Total viable count (CFU/mL)	5.85 ± 0.34	4.66 ± 0.13
Total yeast count (CFU/mL)	1.24 ± 0.14	3.06 ± 0.03

^a^ short-term fermented Godulbaegi kimchi (SGK, fermented for 7 days); ^b^ long-term fermented Godulbaegi kimchi (LGK, fermented for 6 months); ^c^ the values are expressed as mean ± standard deviation (*n* = 3).

**Table 2 foods-11-01020-t002:** The yield of extraction and total phenol content from Godulbaegi kimchi by fermentation duration by diverse solvents.

Extract Solvent	Yield Extraction (%)	Total Phenolic Content (µg of GAEs/mg Extract)
	LGK ^1^	
Ethanol	52.58 ± 2.06 ^b^	77.40 ± 1.42 ^a^
Methanol	61.51 ± 1.21 ^a^	79.01 ± 2.36 ^a^
Water	48.54 ± 0.97 ^c^	38.02 ± 0.80 ^b^

^1^ long-term fermented Godulbaegi kimchi (LGK, fermented for 6 months). Values are mean ± standard deviation (*n* = 3). Means in the same column and with different lower-case letters (^a–c^) indicate significant difference (*p* < 0.05).

**Table 3 foods-11-01020-t003:** The activity of nitrite scavenging (%) of Godulbaegi kimchi by fermentation duration using diverse solvents.

Concentration (%)	pH	Ethanol	Methanol	Water
LGK; Long-term fermented Godulbaegi kimchi	1.2	80.97 ± 0.87 ^aA^	81.90 ± 2.17 ^aA^	74.86 ± 2.05 ^bA^
	3.0	72.30 ± 1.96 ^aB^	72.80 ± 1.50 ^aB^	54.44 ± 2.03 ^bB^
	6.0	26.70 ± 0.96 ^aC^	30.49 ± 2.04 ^aC^	21.32 ± 1.25 ^bC^

Values are mean ± standard deviation (*n* = 3). Means in the same column with different capital case letters (^A–C^) and same row with different lower-case letters (^a–b^) were significantly different (*p* < 0.05).

**Table 4 foods-11-01020-t004:** Comparison of antimicrobial activities of long-term fermented Godulbaegi kimchi.

Strain No.^a^	Negative Control ^b^	*Escherichia coli*	*Salmonella typhimurium*	*Staphylococcus aureus*	*Listeria monocytogenes*
1	- ^c^	+++	++	-	-
2	-	++	-	-	-
3	-	++	-	-	-
4	-	++	++	-	-
5	-	++	-	-	-
6	-	++	++	-	-
7	-	++	-	-	-
8	-	++	-	-	-
9	-	++	++	-	-

^a^*Escherichia coli* KCCM 21052; *Salmonella typhimurium* P99; *Staphylococcus aureus* KVCC BA1100335; *Listeria monocytogenes* KVCC BA0001449. ^b^ Each of the extraction solvent. ^c^ Degree of clarity of clear zone by growth inhibition: +++: Strong inhibition (≥15 mm), ++: clear inhibition (≥12 mm, <15 mm), +: slight inhibition (<10 mm), -: No inhibition.

## Data Availability

Data is contained within the article.

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
