# Peer review of "Effect of Fermentation Duration on the Quality Changes of Godulbaegi Kimchi"

_foods, 2022, doi:10.3390/foods11071020_

Round 1

Reviewer 1 Report

The authors determined the antioxidant activities of godulbaegi kimchi at long-term fermented godulbaegi kimchi (LGK; > 6 months fermentation), number of lactic acid bacteria, total bacteria and yeast, antimicrobial activity of some selected strains and identification of some lactobacillus strains by PCR.

The entire manuscript proves significant linguistic and scientific coherence gaps from the outset. The abstract requires major correction and concentration of more accurate information. The introduction is a sum of general enumerations, and that is all. The entire literature study requires a resumption of information and a focus on the chosen topic of the manuscript,  as the present manuscript has only general information, which only offers volume, but it does not coherently emphasize anything.

 Poor English only accentuates this fact.

Materials and methods

Table 1 does not make sense as long as the recipe appears in text 103-115.

The numbering of the subchapters in the manuscript is done carelessly: 2.1 is presented twice, and so forth.

130 in text: Several lactic acid bacteria (LAB), total bacteria, and yeast were determined. What were the criteria for the selection of bacteria?

What were the standards used for these methods? I suspect you used ISO 4833-2:2013; Microbiology of the Food Chain—Horizontal Method for Enumeration. Microorganisms, in which case you got the temperature wrong.

Concerning the yeast count, did you consider the water's activity?   This often influences the result and the type of yeast that grows, helping to choose the culture medium, be it DG-18 or DRBC. PDA is a   culture medium less used in laboratories that work on food safety.

For what reasons have you chosen the strains tested for antimicrobial activity? Are they identified? What does KVCC BA0001449 mean? There are no  ATCC strains, so please explain. Why was the testing done on only one concentration, and how and why was it selected?

What is the purpose or meaning of PCR in this paper?     What is the resulting information?

Results and Discussion

The limited discussions here give more a sense of results and just that, with no correlation to the literature.

There is no correlation between Materials and methods regarding numbering and Results and discussions.

At 3.2. - Microbial analysis, from the point of view of the total number of germs, the variations are minor,  their importance is exaggerated.

There is no discussion about it at all.

3.3:   appears in the results, but in materials and methods, is completely missing the method of preparation of extracts, being listed only solvents, without other detail.

The T-test done at 3.3 is meaningless.

The discussions at 3.4 are again general without or without validating/supporting the results obtained by the authors.

The results obtained at 3.5 do not in any way suggest that it might be a potential candidate for reducing oxidative stress-associated diseases in humans, the data obtained being minimum in this sens.

The importance of the preliminary results in this study is exaggerated.

Lactobacillus Brevis LAP2 has not been confirmed in this study, so the citation is unusable.

I suggest a further reanalysis of the manuscript and considering a better presentation of results or, in the case of 3.8, better usage of data regarding discussions and interpretation.

Author Response

Thank you for your consideration. I revised all manuscripts based on your comment. The abstract and introduction have been revised to mitigate these concerns.

Thank you, Sincerely.

Reviewer 2 Report

The work is focused on a known food product only in Asia. Then, What could be the relevance of lactic acid bacteria in similar products produced in regions different from Asia? 
L93 Please previous to use the acronym LAB,  defined. 
The methodology is well described. Based on the primary formulation to produce kimchi, what could be the effect of the main ingredients such as onions, ginger, shrimp, garlic, among others?
L122-123: It is essential to give more details about short-term and long-term terms. It is ambiguous. 
Section 2.2 Please add a reference to the assay employed. 
According to the analysis description, it seems that the authors evaluated only the product produced during long-term fermentation. The analysis definition did not agree with the title, main idea, and results. 
Tables. It is essential to add more information, as the mean comes from how many samples or determination. 
The results were described but not discussed. Lake of good explanation of the effect of the fermentation period (one week/ six months).
Why were only the nitrite scavenging and antimicrobial activity evaluated on the long-term fermented product?
What is the recommendation of the authors? Based on the obtained results, long-term or short-term fermentation? 

Author Response

Thank you for your detailed opinion. I revised all manuscripts based on your comment. 

Again thank you, Sincerely.

Reviewer 3 Report

The paper showed a simple change of pH, microorganism and phenolic substances in the fermentation of kimchi , and then analyze the antioxidant ability, microorganism diversity. However, the author analyzes the phenomenon, but there is no complete conclusion and insufficient discussion. In particular, there is no relationship between microorganisms and antioxidation, but to analyze the antibacterial properties of microorganisms. The overall experimental design lacks logic.

Author Response

Thank you for your good focused opinion. To solve this comment, I worried a lot. I revised manuscripts based on your comment. Again thank you, Sincerely.

Reviewer 4 Report

Review on manuscript: foods-1627205

Antioxidant Activity of Godulbaegi Kimchi according to the Duration of Fermentation

by Jung-Min Park, Bo-Zheng Zhang, and Jin-Man Kim

submitted to Foods

In the manuscript submitted for evaluation, the authors studied the some properties of godulbaegi kimchi during fermentation.

Due to the pro-health properties of fermented food, the topic discussed by the authors is interesting and timely, but the way it is presented and the lack of general conclusions indicate the local nature of the results. Moreover, the manuscript requires corrections and supplements in many places.

Detailed recommendation:

Title – does not fully reflect the content of the work,

lines 15-16 – authors should indicate that extracts based on various solvents were analyzed, otherwise the statement is unclear,

lines 104-106 – the manufacturers and origin of the materials used should be specified,

line 109 – drying conditions should be given,

lines 109-111 – if the detailed composition / recipe is given in the table, there is no point in repeating it in the text,

line 113 – GK or SGK? how long did this sample ferment?

line 115 – the wording "> 6 months" is imprecise,

lines 124-125 – concentration should be expressed in moles,

lines 130-137 – the methods used are briefly described and there is no reference to the literature, it does not give the possibility of repeating the experiment,

lines from 139 – the origin of all reagents should be reported,

lines 141 and 144 – condition should be specified,

lines 142-143 – what does "and water (1:25, w / v)" mean?

line 154 – the concentration range for standard curve should be mentioned,

line 169 – why were the analyzes performed at different pHs? how was the pH controlled?

Table 1 – the meaning of "+" and "-" should be explained in the legend,

lines 246-247 – should be: … ethanol extraction> water extraction …

Table 2 – the explanatory indices a and b should be replaced, as they also indicate the significance of differences between the means,

line 292 – Italic style should be used,

Figure 1 – the methodology lacks information that the analyzes were performed for different concentrations,

lines 304-319 – here is the same information as on lines 256-271,

line 323 – why it was in such pH values that the analyzes were performed? it is practically only an acid environment,

lines 325-326 – Table 3 shows that there is no significant difference between the methanol and ethanol extract,

lines 345-346, 365 and Table 4 – such numbering of samples is unclear without further explanation,

line 387 – there are no conclusions,

References – should be reduced.

Author Response

Thank you for your sincere opinion.

I revised manuscripts based on your comment. Again thank you.

Round 2

Reviewer 1 Report

Thank you for taking my suggestions into account. I believe it brings added value to your manuscript.

Reviewer 4 Report

After reassessing the manuscript foods-1627205 currently entitled "Effect of Fermentation Duration on the Quality changes of Godulbaegi Kimchi" by Jung-Min Park, Bo-Zheng Zhang, and Jin-Man Kim, I can state that authors made necessary corrections taking under consideration mostly of my recommendations.